# Walkable Cities: Using the Smart Pedestrian Net Method for Evaluating a Pedestrian Network in Guimarães, Portugal

**Fernando Fonseca** [ID], **Escolástica Fernandes and Rui Ramos** *[ID]

CTAC (Centre for Territory, Environment and Construction), University of Minho, 4800-058 Guimarães, Portugal
* Correspondence: rui.ramos@civil.uminho.pt; Tel.: +351-253-510-200

**Abstract:** Evidence for the benefits of walking has attracted the attention of researchers and practitioners and encouraged them to develop healthier and more sustainable walkable cities. Many methods and approaches have been developed to measure walkability; namely, by using land use attributes. This paper examines the transferability of the Geographic Information System (GIS) based multi-criteria method developed in the Smart Pedestrian Net (SPN) research project to evaluate the level of walkability in a pedestrian network in Guimarães, Portugal. The method involves the assessment of 19 built environment and streetscape attributes, which were scored by a group of experts following the analytic hierarchy process. The method proved to be efficient in evaluating the pedestrian network and in mapping walkability in the study area. Around 65% of the street lengths scored above 0.60, indicating that the overall pedestrian conditions are favourable, with the best performance criteria being those related to accessibility and street connectivity. The method also allowed for the identification of different levels of walkability within the study area and the lack of a pedestrian network of highly scored streets. According to the results, the SPN method could be replicated in other cities to evaluate walkability and could be a useful planning tool to support policies towards developing more walkable cities.

**Keywords:** active mobility; pedestrian activity; walkability; built environment attributes; streetscape attributes; multi-criteria analysis; analytic hierarchy process

## 1. Introduction

As almost every trip begins and ends with walking on a sidewalk, pedestrian infrastructure is a key component of any sustainable transport system. Pedestrian infrastructure is considered the primary means of access to public spaces and the most complex transport network as it joins together all transport modes [1,2].

A pedestrian network consists of all interconnected path segments of the pedestrian infrastructure that can be mainly or exclusively used for people travelling on foot or on tiny wheels [3,4]. This includes not only sidewalks, but also all formal and informal paths that pedestrians have legal access to, such as pedestrian-only zones and streets, shared streets, crossings, pedestrian bridges and tunnels, stairs and ramps, short-cuts, and trails in parks and open spaces, among others [5]. Pedestrian networks have been mostly studied to analyse the levels of connectivity and accessibility in urban spaces, but also to support the development of pedestrian navigation systems [6–8]. Previous research has shown that a suitable pedestrian infrastructure improves the comfort and safety of walking and encourages people to walk [9].

Pedestrian networks have been mainly represented as topological maps that contain the geometric relationships between all path segments [7,10]. Topological maps can provide some important clues about the level of street integration, and the directness and availability of alternative routes between destinations, which have a widely recognised influence on pedestrian accessibility and on travel behaviour [11]. Besides some recent efforts to represent pedestrian vertical movements in 3D through stairs and elevators [3], pedestrian

network maps have been mostly represented in a 2D graphic way, where the edges are sidewalks and the nodes are intersections or where the sidewalks meet end-to-end [12].

One of the main problems in studying pedestrian networks is the lack of pedestrian infrastructure data [7,13]. For that reason, street network data have often been used as a proxy for pedestrian networks by assuming that sidewalks are along the streets [3]. While in some cases, street networks have been found to work well as a proxy for pedestrian networks [1], in others the use of street network data reduces the level of accessibility and connectivity as it contains fewer connections than the real pedestrian network [1,14,15], which in turn reduces the overall walkability [16]. For these reasons and whenever possible, the use of pedestrian network data should be preferred [17].

However, the overall experience of walking is not only influenced by the characteristics of the pedestrian infrastructure, but also by many other built environment and streetscape attributes. Pedestrians are very sensitive to overall walking environment features, which define the extent to which walking is a convenient mode of transport and the pedestrian routes are comfortable, safe, connected, and attractive to walk [4,17]. A connectivity/accessibility analysis of the pedestrian infrastructure can only give a partial view of the several factors that shape the decision and satisfaction of walking. Thus, other neighbourhood and micro scale street attributes should be considered when studying pedestrian networks.

This paper describes a GIS-based multi-criteria method to evaluate the level of walkability of a pedestrian network in Guimarães, Portugal. This study is motivated by two main goals. The first goal consists of examining the spatial transferability of the method developed in the Smart Pedestrian Net (SPN) research project, which was previously implemented in Bologna and Porto with the goal of promoting walking as one of the critical dimensions of smart and sustainable mobility in European cities. As highlighted by Lefebvre-Ropars et al. [18], assessing the spatial transferability of measures to evaluate walkability is a critical step for research and practice as they may validate a given construct for use in transport planning and modelling applications. The second goal consists of evaluating the level of walkability in the pedestrian network in the centre of Guimarães. The evaluation includes 19 built environment and streetscape attributes related to the following six criteria: accessibility, land use, connectivity, sidewalk facilities, safety/security, and streetscape design. This study details how the data of the 19 attributes were collected and analysed, how the various criteria and sub-criteria were weighted, and the GIS operations necessary to rank the streets and generate a pedestrian network walkability map. This second goal also intends to demonstrate that evaluating the walkability of a pedestrian network gives a more complete overview than merely evaluating street networks based on accessibility and connectivity. In addition, it also intends to rectify the lack of studies combining both neighbourhood and micro-scale environmental factors, which is a problem recently reported by Molina-García et al. [19]. In the author's view, this study is an important methodological step towards the wider use of the SPN on walkability research. Moreover, the results can be used by planners and decision-makers to support policies towards developing more walkable cities.

This paper is organised as follows. The next section presents a review on the main criteria that influence walkability in pedestrian networks. Then, Section 3 describes the method and data adopted in this study. The results are presented in Section 4 and discussed in Section 5. Finally, the last section summarises the main conclusions of this study.

## 2. Related Work

Developing walkable cities is a way to create affordable and equitable transport systems for the entire urban community. The built environment is the physical support of all activities, services, and infrastructures found in urban spaces [20]. The extent to which the built environment is pedestrian friendly and enables walking has been widely defined as walkability. In a recent review, Fonseca et al. [21] showed that many different methodologies have been used to measure the influence of the built environment and streetscape

attributes on walkability. This includes walkability indexes based on a changeable number of attributes, qualitative approaches based on questionnaires and surveys, audit tools, inventories and web-based services, such as Walk Score. One of the problems arising from the profusion of methods is the dispersion and unclear structuring of the evaluation methods to describe and measure walkability.

In recent years, solid contributions have been made to categorise the influence of built environmental attributes on walkability. The most relevant include: (i) survey tools such as the NEWS (Neighbourhood Environment Walkability Survey) [22], which covers pedestrian infrastructure data as well as residential density, land use mix, land use mix access, street connectivity, traffic safety, security from crime, and aesthetics data; (ii) the GIS walkability index developed by Frank et al. [23], which has been highly replicated and adapted as a composite measure of land use mix, street connectivity, and residential density; (iii) the 5D layout of Ewing and Cervero [24], which includes density, diversity, design, destination accessibility, and distance to public transport as critical drivers of travel behaviour; and (iv) the 5C layout developed by the Greater London Authority, which defines connectivity, convenience, comfortability, conviviality, and conspicuousness as critical dimensions to allow pedestrians to walk with high-quality levels [25,26]. Based on these referential works, it can be argued that a pedestrian network should be convenient, comfortable, connected, safe, and attractive.

To be convenient, a pedestrian network should enable walking as an alternative mode of transport to access daily goods and services [25,27]. This dimension comprises attributes related to land use diversity and density. Having compact urban structures with mixed land uses (residential, services, retail, recreational) typically reduces the distances need to travel, making trips on foot more convenient [28]. Proximity to diverse facilities such as public transport, schools, retail, and parks is critical for pedestrians. As a rule of thumb, compact urban environments where destinations are within 10 and up to 20 min walking distance are more convenient [29]. Previous research has consistently shown that areas with diverse land uses and short distances to destinations are more conducive for walking [23,30–32]. Areas with high residential densities are usually characterised by more pedestrian activity [23,33]. They usually attract services and retail, which helps to reduce the walking distances to these destinations.

To be comfortable, a pedestrian network should make the walking experience pleasant and suitable [25,34]. Comfort is related to pedestrians' emotional reactions, and this dimension is mostly influenced by pedestrian infrastructure attributes. This includes variables such as the characteristics and condition of sidewalks and remaining pedestrian infrastructure, the presence of obstacles on sidewalks, street trees, street furniture, and slopes, among others. These attributes have been insufficiently included in walkability indexes due to the lack of this type of micro data [35]. Nonetheless, research has shown that well designed and maintained pedestrian infrastructure is critical for pedestrians. For example, the presence of wide, well-maintained, and clean sidewalks has been reported as enabling comfortable walking experiences [36–38]. Similarly, sidewalks without physical obstacles, such as parked cars, and with street furniture, are also more comfortable for pedestrians [35]. Sidewalks with street trees providing shade are also described as being more thermally comfortable [34,39].

To be connected, a pedestrian network should link key origins, such as residential areas, to key destinations, such as transport hubs and city centres. This means that origins and destinations should be connected by continuous pedestrian infrastructure without interruptions and obstructions [26]. In addition, more interconnected streets provide more alternative routes, which reduces distances and makes walking more convenient [40]. Connectivity has been mostly analysed as a meso urban design variable, by using a multitude of attributes associated with the street layout. From these, the most used have been intersection and street density due to the availability of such data in GIS format [21]. Other authors have examined the topological distance, i.e., the number of turns that are needed to reach one location from another in a network. Previous studies showed that pedestrians

prefer routes with few directional changes [41]. Areas providing high street connectivity have been correlated with more walking and physical activity [23,42,43].

To be safe, a pedestrian network should provide traffic protection, as pedestrians are vulnerable road users in the case of collisions, as well as personal security, so that pedestrians can walk without being afraid of incivilities and crime. This dimension has been analysed by considering different street level attributes. In the case of traffic safety, these include traffic volume, traffic speed, traffic lanes, risk of accidents, and traffic calming devices, among others. In the case of public security, these attributes include graffiti on buildings and structures, vacant buildings, deteriorating buildings, street lighting, homicide rates, police stations, and pedestrian activity, among others [21]. In general, a lack of traffic safety and public security has been reported as a main barrier deterring people to walk [35,42,43]. In the case of traffic safety, there is some consistency with the fact that pedestrians prefer quiet streets with low traffic speeds and volumes [38] and streets with few lanes to cross [44]. The influence of security on walking is more evident in cities characterised by urban violence. In these cases, street lighting, surveillance systems, and police stations are known to enhance the perception of security [45].

Finally, to be attractive, pedestrian networks should provide inviting and pleasant conditions for pedestrians. Several street-level qualities define the extent to which walking routes and publics spaces are attractive for pedestrians, such as: complexity, indicating the visual richness of a place in terms of building shapes, styles, colours, and furniture [46]; human scale, reflecting how buildings and spaces are scaled to human size and needs [47]; level of enclosure, showing how enclosed by vertical elements the spaces are [46]; and visual transparency, showing the degree to which people can see or perceive human activity through windows and doors [46]. Aesthetic and street design data are difficult to collect and measure, which explains the relatively low use of these data in walkability studies [48]. However, street design qualities are known for having an overall positive impact on walking [47,49].

From this brief background, it can be concluded that a high-quality pedestrian network should include a set of factors and attributes that go beyond a merely topological map representing sidewalks and intersections. It is necessary to take into account the surrounding urban environment, including the built and streetscape attributes, which also contribute to making a pedestrian network more or less walkable.

## 3. Method and Data

### 3.1. Case Study

The city centre of Guimarães was selected as the area in which to conduct the case study. Guimarães is a medium-sized Portuguese city located in the Northern Portugal that has around 52,000 inhabitants. In the 12th century, Guimarães became the first capital of Portugal. The historic centre of the city (Figure 1) was included in the UNESCO World Heritage List in 2001. The city is recognised as a valuable and well-preserved heritage site, containing building typologies that exemplify the development of Portuguese architecture from the 15th to the 19th centuries. During the last thirty years, policies have focused on the rehabilitation of public spaces and buildings and on the adaptive reuse of previous industrial space as housing and office space for arts, innovation, and research services [50]. Policies have been focused on making the city more sustainable and more liveable. This includes, for example, the candidacy of Guimarães to the European Green Capital Award 2020 [51] and the Sustainable Urban Mobility Plan (SUMP), (SUMP), to be implemented until 2027, which has ambitious goals. Encouraging the use of the pedestrian mode is one of the strategic objectives of the SUMP. This will be undertaken by increasing the number of pedestrian areas and pedestrian routes (particularly to schools), by restricting car access to some parts of the city, and by improving traffic safety at crossing points with waiting/walking times and audible signals [51]. For these reasons, the evaluation of the walkable conditions in Guimarães is particularly interesting and appealing.

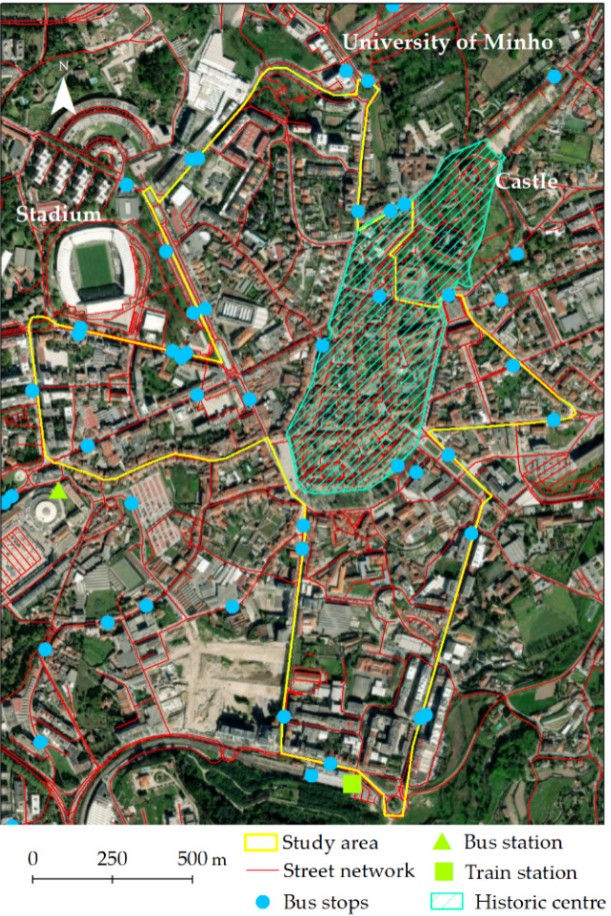

**Figure 1.** Study area in the city centre of Guimarães.

A sector in the centre of Guimarães (Figure 1) was selected to examine the spatial transferability of the SPN method and to evaluate the level of walkability in this pedestrian network. The selected area, which includes part of the UNESCO World Heritage Centre, is 1.6 km² and has an average distance from the central point to the limits ranging from 500 to 1400 m. This area presents a compact urban structure, with a varied sample of spaces and streets with different land uses and levels of occupation. This includes residential areas, retail, public and private services, green spaces, and cultural and tourist destinations. In the fringes, there are important amenities and pedestrian generation poles (a university, a stadium, a hospital, hypermarkets, bus and train stations, the castle, etc.) that make the evaluation of the pedestrian conditions in this area particularly interesting. The study area contains 90 streets divided into 118 paths, squares, and footpaths, which sum a total length of 27.8 km.

As suggested by Zhang and Zhang [17], we used not only street network data, but also pedestrian network data, including pedestrian-only streets, footpaths, and squares. In terms of hierarchy, 3% of these streets are arterial, 18% are collector, and 79% are local access streets, including dead-end streets, shared streets, pedestrian-only streets, and footpaths.

### 3.2. Multi-Criteria Evaluation

Multi-criteria evaluations (MCAs) have been widely used in walkability studies [26,40,52]. The MCA adopted in this study was developed considering two main phases: (i) selecting and evaluating the attributes; and (ii) weighting the respective attributes. The steps performed in these two phases are described in the following subsections.

### 3.2.1. Selecting and Evaluating the Attributes

The 19 built environment and streetscape attributes used in the SPN project [53] were chosen to assess the level of walkability in the selected pedestrian network of Guimarães. It should be emphasised that these attributes resulted from an exhaustive literature review on the influence of built environment and streetscape attributes on walkability [21]. In the current study, these 19 attributes were evaluated as sub-criteria of the following six criteria: accessibility, land use, street connectivity, sidewalk facilities, safety and security, and streetscape design, which contribute to making a pedestrian network convenient, comfortable, connected, safe, and attractive (Figure 2). The listed attributes and the rationale for their usage, and well as the evaluation methods and the respective data sources are described below, while a summary of this information can be found in Table 1.

**Table 1.** Criteria and sub-criteria proposed and respective evaluating measures.

| Criteria | Sub-Criteria | Measure | Source |
|---|---|---|---|
| Accessibility | Proximity to public transport | Bus stops $\leq$400 m = 1 >400 m = 0; Train stations $\leq$800 m = 1 >800 m = 0 | GIS |
| | Proximity to car parking | Car parking $\leq$500 m = 1 >500 m = 0 | GIS |
| | Proximity to community facilities | Educational $\leq$800 m = 1 >800 m = 0; Health $\leq$800 m = 1 >800 m = 0; Cultural $\leq$400 m = 1 >400 m = 0; Public services $\leq$400 m = 1 >400 m = 0; Recreational $\leq$1000 m = 1 >1000 m = 0; Religious $\leq$500 m = 1 >500 m = 0 | GIS |
| Land use | Land use mix | >3 uses = 1 $\leq$3 uses = 0 | GIS |
| | Residential density | $\geq$average density = 1 <average = 0 | GIS |
| | Retail density | $\geq$average density = 1 <average = 0 | GIS |
| Sidewalk facilities | Sidewalk width | $\geq$1.5 m = 1 <1.5 m = 0 | Audit |
| | Sidewalk condition | Good = 1 bad = 0 | Audit |
| | Obstacles on sidewalks | No = 1 Yes = 0 | Audit |
| | Trees on sidewalks | Yes = 1 No = 0 | Audit |
| | Slopes | $\leq$5% = 1 >5% = 0 | GIS |
| | Street furniture | Presence of 4 elements = 1 No (<4) = 0 | Audit |
| Street connectivity | Intersection density | $\geq$3 intersections = 1 <3 intersections = 0 | GIS |
| Traffic safety and security | Traffic speed | $\leq$30 km/h = 1 >30 km/h = 0 | Audit |
| | Traffic lanes | $\leq$2 lanes = 1 >2 lanes = 0 | Audit |
| | Pedestrian activity | Pedestrian activity = 1 no pedestrian activity = 0 | Audit |
| Streetscape design | Enclosure | H/W ratios of 1:2 to 1:4 = 1 H/W ratios >1:2 and <1:4 = 0 | Audit |
| | Complexity | Variation of 4 elements = 1 No (<4) = 0 | Audit |
| | Transparency | $\geq$50% transparency = 1 <50% = 0 | Audit |

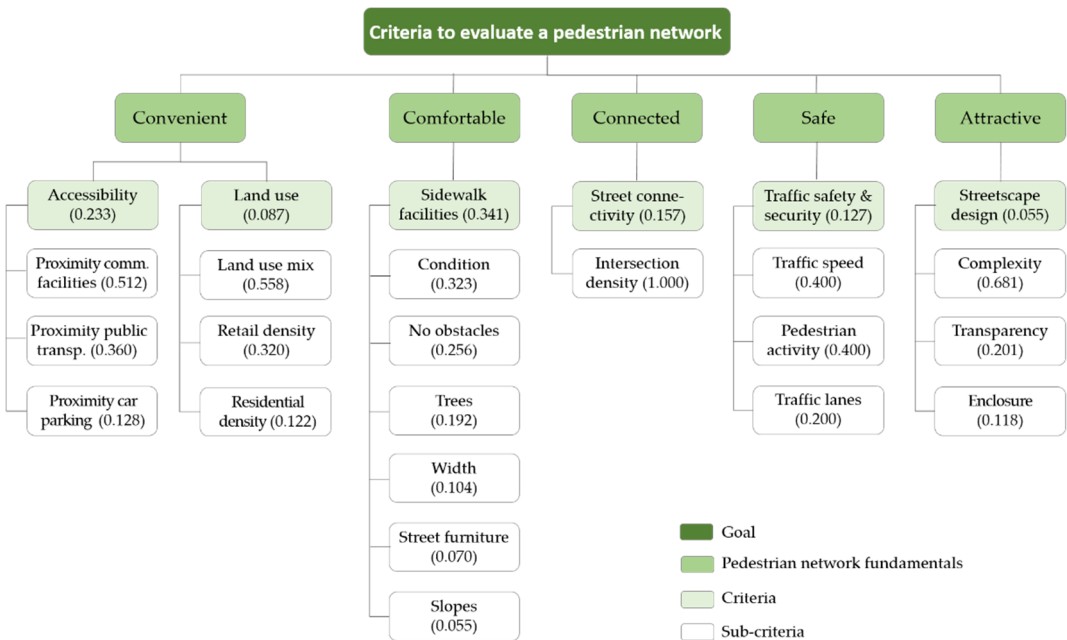

**Figure 2.** AHP decision tree and weights assigned to the criteria and sub-criteria.

- Proximity to public transport: stops and stations within adequate distances increase the odds of walking to public transport. In general, people are usually willing to walk around 400 m to bus stops and 800 m to rail stations [40]. This attribute was assessed through GIS Euclidian buffers of 400 m from bus stations and of 800 m from the train station. The binary evaluation used was: streets/segments within the buffers scored 1, otherwise 0. Public transport data were retrieved from GCC [54].

- Proximity to car parking: the decision to drive decreases with increasing walking distance to car parking, but distances to car parking are usually shorter than to public transport stops [55]. According to Fonseca et al. [56], on average, people are willing to walk 500 m to car parking. This attribute was assessed through GIS Euclidian buffers of 500 m from car parking. The binary evaluation used was: streets/segments within the buffers scored 1, otherwise 0. Car parking data were retrieved from GCC [54].

- Proximity to community facilities: facilities within appropriate distances increase the odds of walking rather than driving [29]. This attribute was assessed through GIS Euclidian buffers according to the threshold distances defined in Fonseca et al. [56] to six types of facilities (educational, health, cultural, other public services, recreational, and religious). The binary evaluation used was: streets/segments within the buffers scored 1, otherwise 0. Community facility data were retrieved from GCC [54].

- Land use mix: diverse land use reduces the distance needed to travel, making trips on foot more convenient [28,29]. This attribute was assessed by considering the presence of five land uses: residential, retail, service, institutional, and recreational. The binary evaluation used was: streets/segments with >3 uses score 1, otherwise 0. Land use data were extracted from the points of interest of OpenStreetMap (https://www.openstreetmap.org (accessed on 4 May 2022)).

- Residential density: high residential density areas usually have more pedestrian activity [23,33]. This attribute was assessed as the number of inhabitants per area (inhab./ha) at the census block level. The binary evaluation used was: streets/segments having densities above or equal to the average of the study area scored 1, otherwise 0. The data source was the 2011 Census [57].

- Retail density: high commercial density areas/streets are more attractive for pedestrians as they fulfil various daily needs [58]. This attribute was assessed as the number of retail stores per area (stores/ha) at the census block level. The binary evaluation used was: streets/segments having retail densities above or equal to the average of the

study area scored 1, otherwise 0. Retail store data were retrieved from OpenStreetMap (https://www.openstreetmap.org (accessed on 4 May 2022)). All types of retail stores available were considered.

- Intersection density: more interconnected streets provide more alternative routes, reducing walking distances to destinations [40]. This attribute was assessed as the number of street intersections with ≥3 legs [59]. The binary evaluation used was: streets/segments with ≥3 intersections scored 1, otherwise 0. Street data were retrieved from GCC [54].
- Sidewalk width: sidewalks should have a minimum width of 1.5 m to allow two people to walk side by side, while any width of less than 1.5 m does not meet the minimum requirements for people with disabilities [60]. This attribute was assessed by considering this referential minimum width of 1.5 m at the full length of each street/segment. The binary evaluation used was: sidewalks wider than or equal to 1.5 m scored 1, otherwise 0. Sidewalk data were obtained from street auditing.
- Sidewalk condition: the state of the sidewalks' pavement has a significant influence on pedestrian comfort and on sidewalk accidents and falls [9]. This attribute was assessed by considering the presence of deformities that may cause trip hazards (cracks, holes, raised pavements, etc.) at the full length of each street/segment. The binary evaluation used was: sidewalks in good condition, e.g., without major deformities, scored 1, otherwise 0. Sidewalk data were obtained from street auditing.
- Obstacles on sidewalks: sidewalks without physical obstructions provide more comfortable walking experiences [35]. This attribute was assessed by considering the presence of permanent and temporary obstacles on sidewalks (bus stops, furniture, parked cars, cafe tables, etc.) that reduce their width to <1.5 m. The binary evaluation used was: sidewalks without obstacles scored 1, otherwise 0. Sidewalk data were obtained from street auditing.
- Trees on sidewalks: the presence of trees makes sidewalks more comfortable and pleasant to walk on [34,37]. This attribute was assessed by considering the presence of trees on both sides of the sidewalks in segments of at least 10 m. The binary evaluation used was: sidewalks with trees scored 1, otherwise 0. Sidewalk data were obtained from street auditing.
- Slopes: slopes have a significant influence on walking as they affect travel speeds and the effort required to walk [39]. This attribute was assessed by considering the threshold slope of 5%, above which the slope is considered unattractive for pedestrians [61]. The binary evaluation used was: sidewalks with slopes ≤5% scored 1, otherwise 0. Sidewalk elevation data were retrieved from Google Earth by estimating the difference in elevation between the endpoints of the streets/segments divided by the difference in distance between them.
- Street furniture: the presence of street furniture makes pedestrian environments more comfortable and attractive [62]. This attribute was assessed by considering the presence of the following four items: benches, litterbins, streetlamps, and pedestrian signage. The binary evaluation used was: sidewalks with the four items scored 1, otherwise 0. Street furniture data were obtained from street auditing.
- Traffic speed: high vehicle speeds substantially increase the risk of injury and death to pedestrians and, for that reason, pedestrians prefer quiet streets with low traffic speeds [38]. This attribute was assessed by considering the traffic speed limit of 30 km/h, which is considered safer for pedestrians [63]. The binary evaluation used was: low speed streets (≤30 km/h) scored 1, otherwise 0. Traffic speed data were obtained from street auditing.
- Traffic lanes: multi-lane streets are more difficult to cross, increasing the risk of accidents, and for that reason, pedestrians prefer to cross streets with no more than two lanes [44]. This attribute was assessed by considering the number of traffic lanes at each street/segment. The binary evaluation used was: streets with ≤2 lanes scored 1, otherwise 0. Traffic lanes data were obtained from street auditing.

- Pedestrian activity: "more eyes" on the street creates an informal surveillance system that improves the sense of security [35]. This attribute was assessed by considering the number of pedestrians observed during the street auditing [64]. The binary evaluation used was: no pedestrians/no signs of pedestrian activity scored 0; some/high pedestrian activity scored 1.
- Enclosure: a well-enclosed streetscape increases the perception of intimacy and security [65]. This attribute was assessed by considering the height-to-width ratio (H/W), e.g., the relation between vertical elements (buildings and trees) and the horizontal space between them (street width). The binary evaluation used was: streets with H/W ratios between 1:2 and 1:4 scored 1, otherwise 0. Enclosure data were obtained from street auditing.
- Complexity: this reflects the visual richness of a place and has been described as a design quality significantly correlated with walkability [66]. This attribute was assessed by considering the diversity of building colours, architectural styles, and the presence of outdoor dining and public art [46]. The binary evaluation used was: streets with the variations of the described elements scored 1, otherwise 0. Design complexity data were obtained from street auditing.
- Transparency: streets providing visual transparency have been significantly correlated to walkability [67]. This attribute was assessed as the proportion of transparent windows/doors at the street level [46]. The binary evaluation used was: streets with ≥50% of their length providing high transparency, scored 1, otherwise 0. Visual transparency data were obtained from street auditing.

### 3.2.2. Weighting the Attributes

Weighting is a critical aspect of multi-criteria decision analysis. In the SPN project, the 19 criteria were weighted by taking into consideration the opinion of 1438 individuals surveyed in Bologna and Porto [53]. In the SPN, the weights were derived from the mean of a 5-point Likert scale through which the attributes were evaluated. In order to make the evaluation more connected with the conditions found in Guimarães, we decided to support the weighting process with a panel of experts from the University of Minho using the analytic hierarchy process (AHP). Developed by Saaty [68], the AHP has been extensively used in spatial multi-criteria decision analysis for solving different problems among complex criteria, which are usually in interaction [69,70]. The process was conducted according to four main steps: (i) defining the hierarchical map; (ii) expert participation; (iii) pairwise comparison; and (iv) consistency checking.

The first step consisted of defining the hierarchy index system to divide the decision problem into a hierarchy map where the various criteria and sub-criteria were presented. The obtained map is shown in Figure 2.

In the second step, the 19 attributes were evaluated by a total of 28 experts from the University of Minho, including professors, researchers, and PhD students mainly from the Schools of Engineering, Architecture, and Geography, which are located in Guimarães. They participated by completing an online questionnaire in 2019.

The third step involved the pairwise comparison among the various attributes selected to solve the problem. The pairwise comparison was used to convert the importance of effecting factors' judgements into numerical values and for defining the priorities of the attributes. The pairwise comparison can be represented in the form of a matrix with a size of $n \times n$, where $n$ is the number of criteria. The pairwise comparison matrix can be represented as shown in Table 2.

**Table 2.** AHP pairwise comparison matrix.

|  |  | $A_1$ | $A_2$ | . . . | $A_n$ |
|---|---|---|---|---|---|
|  | $A_1$ | 1 | $a_{12}$ | . . . | $a_{1n}$ |
| $A = a_{ij} =$ | $A_2$ | $1/a_{12}$ | 1 | . . . | $a_{2n}$ |
|  | . . . | . . . | . . . | . . . | . . . |
|  | $A_n$ | $1/a_{1n}$ | $1/a_{2n}$ | . . . | 1 |

With such comparison, the so-called dominance coefficient ($a_{ij}$) is obtained, which represents the relative importance of the component on row (*i*) over the component on column (*j*). The comparison consists of assigning values between each pair of criteria to define how each attribute is more or less important than the others [71]. The widely used Saaty fundamental scale [68], which ranges from 1 to 9, indicating equal importance to extreme importance, respectively, was adopted to assign these values. In such a matrix, the upper triangle corresponds to the pairwise comparison (values assigned), the diagonal between the rows and the columns corresponds to the same attributes (value of 1) and the lower triangle is the reciprocal of the upper triangle. For assigning to the *n* elements $A_1$, $A_2$, . . . $A_n$ a set of numerical weights $W_1$, $W_2$, . . . $W_n$ that reflect the judgment made by the experts can be estimated as shown in Table 3.

**Table 3.** AHP weighting process.

|  | $A_1$ | $W_1/W_1$ | $W_1/W_2$ | . . . | $W_1/W_n$ |
|---|---|---|---|---|---|
|  | $A_2$ | $W_2/W_1$ | $W_2/W_2$ | . . . | $W_2/W_n$ |
| $A =$ | . . . | : | : | : | : |
|  | $A_n$ | $W_n/W_1$ | $W_n/W_2$ | . . . | $W_n/W_n$ |

As seen in previous studies [70–72], the pairwise comparison matrix was determined using the geometric mean resulting from the expert's evaluation. Finally, the results were normalized so that the sum of the weightage of each criteria was equal to 1. The normalization was conducted by dividing each value of the comparison matrix by the sum of the values of the column to which it belongs, e.g., the weights of each criterion (the eigenvector values) resulted from dividing the normalized matrix line values by the respective number of criteria.

In the last step, the consistency of the comparison matrix was checked. According to Saaty (1990) and as adopted in other studies using the AHP [73,74], this test consists of calculating a consistency ratio according to the formula shown in Equation (1).

$$CR = \frac{CI}{RI} \tag{1}$$

where, *RI* is the random consistency index, a parameter set by Saaty [68] according to the number of criteria compared and *CI* is the consistency index which, in turn, was calculated according to Equation (2):

$$CI = \frac{\lambda_{max} - n}{n - 1} \tag{2}$$

where, $\lambda_{max}$ is the dominant eigenvalue of the matrix and *n* is the size of the matrix.

A matrix is considered consistent when the consistency index and the consistency ratio satisfies the recommended value of less than 0.1 as proposed by Saaty [68] and as adopted in other AHP applications [73,74]. In our study, as shown in Table 4, both indexes satisfy the recommended value of less than 0.1 defined by Saaty [68]. This shows that the criteria evaluation performed by the experts was reliable and significant.

**Table 4.** Consistency Index (*CI*) and Consistency Ratio (*CR*) for AHP validation.

| Sub-Criteria and Criteria | Lambda Max | Consistency Index | Random Consistency Index | Consistency Ratio |
|---|---|---|---|---|
| Accessibility | 3.108 | 0.054 | ($n = 3$), $RI = 0.58$ | 0.093 |
| Land use | 3.018 | 0.009 | ($n = 3$), $RI = 0.58$ | 0.016 |
| Sidewalk facilities | 6.186 | 0.037 | ($n = 6$), $RI = 1.24$ | 0.030 |
| Street connectivity | - | - | ($n = 1$), $RI = 0.00$ | - |
| Traffic Safety and Security | 3.001 | 0.005 | ($n = 3$), $RI = 0.58$ | 0.008 |
| Streetscape design | 3.025 | 0.012 | ($n = 3$), $RI = 0.58$ | 0.021 |
| Overall six criteria | 6.178 | 0.036 | ($n = 6$), $RI = 1.24$ | 0.029 |

The obtained weights with the described process are shown in Figure 2. Accordingly, the most highly evaluated criterion was sidewalk facilities (0.341), followed by accessibility (0.233) and street connectivity (0.157). In turn, streetscape design was the criterion with the lowest evaluation (0.055). In Figure 2, the various sub-criteria of each group are displayed from the highest to the lowest scored. For example, complexity was much more relevant (0.681) than the remaining streetscape design sub-criteria, while land use mix (0.558) was much more valued than the other land use sub-criteria.

Finally, the last step of the work consisted of determining the overall walkability index of the pedestrian network. This index was defined by combining the evaluation of each sub-criterion with the normalised weights derived from the AHP. This operation was performed on GIS. The software used was ArcGIS of Esri. Once the criteria and their corresponding scores were defined, the values of the six criteria were aggregated and normalised again. The result was a walkability map ranging from 0 to 1, showing the final classification of all street paths according to the various criteria evaluated.

## 4. Results

The evaluation of the six criteria in the study area is shown in Figure 3 and Table 5, while the maps showing the performance of the 19 sub-criteria can be found in Appendix A. In these maps, the green paths correspond to the streets with the highest scores, while the red paths correspond to those with the lowest scores.

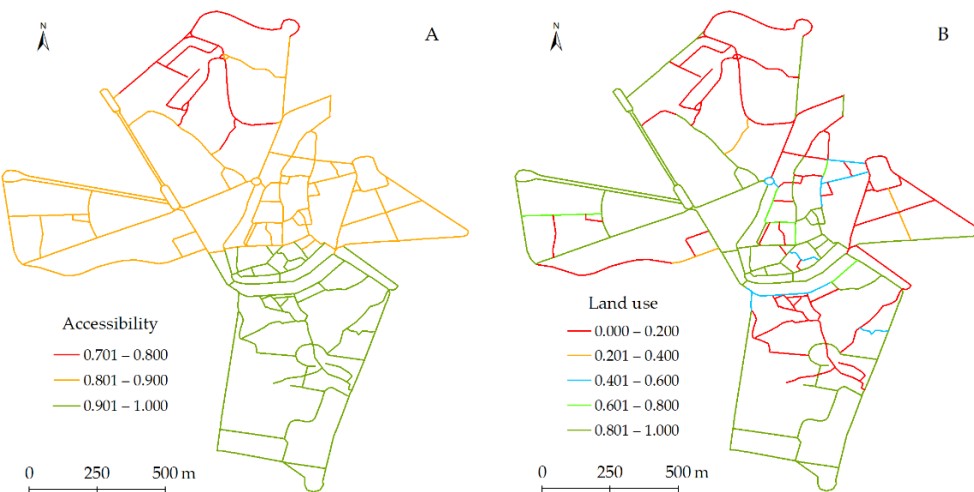

**Figure 3.** *Cont.*

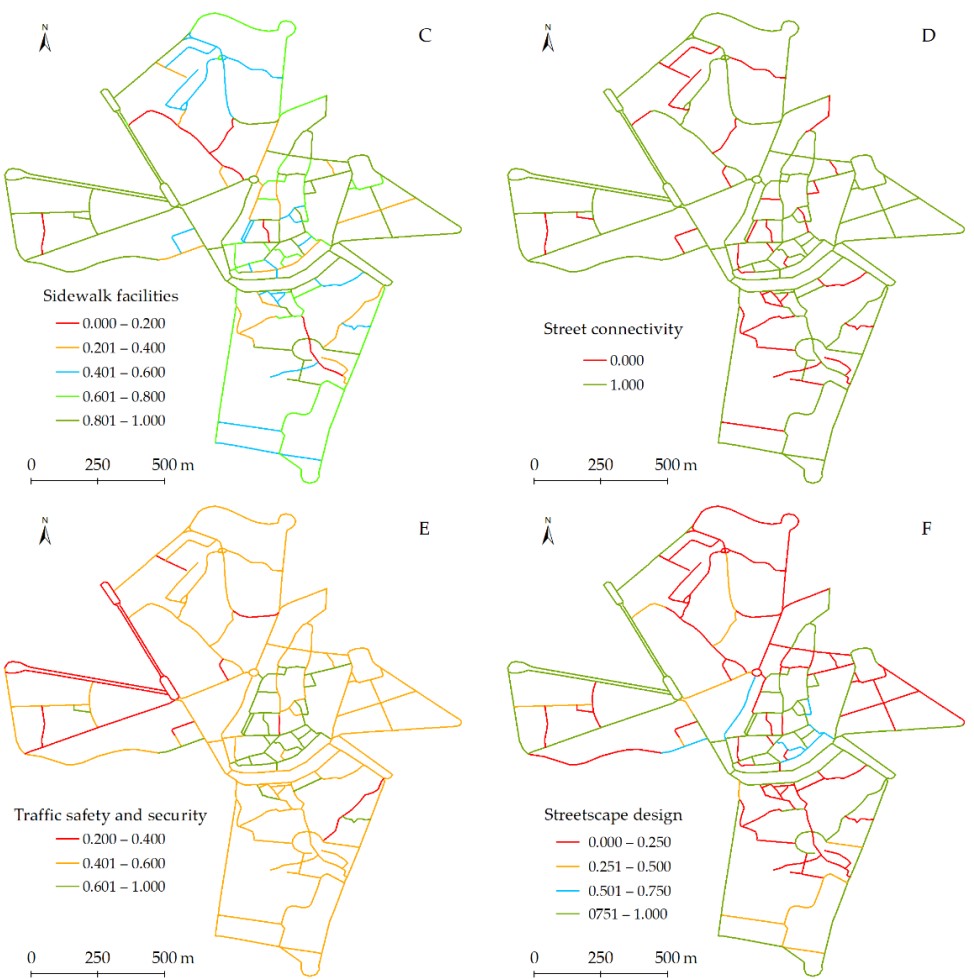

**Figure 3.** Evaluation of the walkable conditions according to the six criteria (**A–F**).

**Table 5.** Main statistics from the evaluation of the six criteria.

| Criteria | Average Evaluation | Max. Evaluation | Min. Evaluation | Street Length (Max. Evaluation) |
|---|---|---|---|---|
| Accessibility (Figure 3A) | 0.88 | 1.00 | 0.72 | 10.29 km |
| Land use (Figure 3B) | 0.51 | 1.00 | 0.00 | 11.31 km |
| Sidewalk facilities (Figure 3C) | 0.69 | 1.00 | 0.06 | 8.59 km |
| Street connectivity (Figure 3D) | 0.74 | 1.00 | 0.00 | 24.99 km |
| Traffic safety and security (Figure 3E) | 0.66 | 1.00 | 0.40 | 4.60 km |
| Streetscape design (Figure 3F) | 0.46 | 1.00 | 0.00 | 9.42 km |

As explained in the Methodology, accessibility was calculated as a measure of distance and proximity to community facilities, public transport, and car parking. With an average performance of 0.88, the level of accessibility to the three considered sub-criteria was largely favourable (Figure 3A). In terms of community facilities, all street paths were within the threshold walking distances to almost all the services considered, including schools, health care and public services, and urban parks. This is explained by the diversity of urban functions and land uses in the centre of Guimarães, which could make walking trips shorter and more convenient. In terms of distance to public transport, the performance is different. All street paths are within 400 m of a bus stop, but only 39% are within 800 m of the train station. As shown in Figure 1, this is explained by the highest density of bus stops (21.3 stops/km$^2$) when compared to the train station (only one station, with a peripheral urban location). This may reduce the propensity of walking to the train station as it may

involve trips longer than 2 km from the north sector of the study area. Finally, all streets paths are within 500 m of a car parking and, therefore, all streets performed positively on this sub-criterion. However, this proximity could encourage car usage, which is a less sustainable and less healthy mode of transport.

In terms of land use, the overall performance of the street paths analysed, which is shown in Figure 3B, was 0.51. The first sub-criterion, land use mix, obtained a classification of 0.54. According to the respective sub-criterion map (Appendix A), the diversity is much higher in the historic centre, rather than in the surrounding spaces. This is because the historic centre is the most preferable location for a variety of multifunctional economic, social, and cultural facilities, which have made these areas urban hubs in which to live, work, trade, and socialise. The second sub-criterion, retail density, had a slightly lower evaluation (0.45) but a similar spatial distribution. Thus, the streets of the historic centre were those performing better due to the presence of various types of retail (clothing and shoe shops, bakeries, kiosks, and convenience stores, among others).

Finally, the residential density obtained an evaluation of 0.54. Interestingly, many streets of the historic centre are amongst those with better performance, which means that the historic centre still has a residential function. The street paths performing better in terms of residential density generally correspond to the paths with higher retail density. This pattern suggests that residential areas with higher densities are attractive for the location of retail shops.

Regarding the analysis by the sidewalk facilities (Figure 3C), the overall performance of this criteria was 0.69, the percentage of paths scoring above this value was 59%, and the length of the streets with the maximum score (1.00) was 10.3 km. The sub-criterion with the best performance was slopes (0.92) due to the overall flatness of this area, while the sub-criteria with the worst performance were the presence of trees (average score of 0.38) and the sidewalk width (0.67). As shown in the respective sub-criteria maps (Appendix A), the lack of trees was mostly detected in the urban core, while the narrower sidewalks were identified in local access and dead-end streets found in some residential sectors and in the city centre. Some of the streets providing better sidewalk facilities can be found around the Stadium, at Avenues S. Gonçalo, Cónego Gaspar, Alameda S. Dâmaso, Alameda Doutor Alfredo Pimenta, among others. This means that the streets with better sidewalk facilities are found on collector streets (avenues) and not on the local access streets of the urban core, where the compact urban structure narrows the sidewalks and restricts the presence of street trees.

With an average evaluation of 0.74, the level of street connectivity in this area was also favourable, which may allow pedestrians to choose between alternative routes to reach their destinations. As shown in Figure 3D, the streets with the worst performance mainly correspond to local access streets (dead-end streets, footpaths) found in various sectors of the study area, including the historic centre.

The overall performance of traffic safety and security is shown in Figure 3E. This criterion obtained an average performance of 0.66, but the three respective sub-criteria performed very differently. The number of traffic lanes had the highest performance (0.93), pedestrian activity performed slightly lower (0.87), while traffic speed obtained a clearly worse evaluation (0.29). In terms of geographic distribution, traffic safety is generally better in the historic centre than in the surrounding neighbourhoods. This is best explained by the compact urban morphology of the urban centre, characterised by streets having just one or two traffic lanes, with low speed limits (30 km/h), and with a considerable pedestrian activity. In turn, some of the surrounding spaces are crossed by arterial and collector streets that provide less protection from motorised traffic due to the high speed limit allowed (50 km/h) and to the presence of more than two traffic lanes in some collector streets. The overall study area was also characterised by regular pedestrian activity, which ensures a sense of informal surveillance from the people walking in this area.

Finally, the evaluation of the streetscape design is shown in Figure 3F and resulted in an average performance of 0.46. Inversely to the above criteria, the streets ranking the

best and worst were spatially widespread, indicating a considerable variety in terms of streetscape design. Complexity and transparency were the design sub-criteria with the lowest evaluation (0.42 and 0.44, respectively). These two sub-criteria generally performed better in the historic centre and in some arterial avenues (S. Gonçalo, Alameda Doutor Alfredo Pimenta, and Largo do Toural, among others) due to the presence of diverse architectural styles and building colours, public art, outdoor dining, and the presence of more active uses. In these areas, the proportion of transparent windows/doors from facades at the street level was higher due to the location of many retail businesses. Enclosure was the design sub-criterion with the highest evaluation (0.69). Street paths in general provided an acceptable level of enclosure. In the historic centre, the sense of enclosed space is given by the prevalence of narrowed streets. In the remaining areas where the streets are wider and the buildings are lower, the presence of trees helps to define an adequate visual enclosure.

To obtain an overall evaluation of the pedestrian conditions in the study area, the normalized values resulting from the evaluation were combined with the weights derived from the AHP. This process was performed in GIS and the results are shown in Figure 4 and in Table 6.

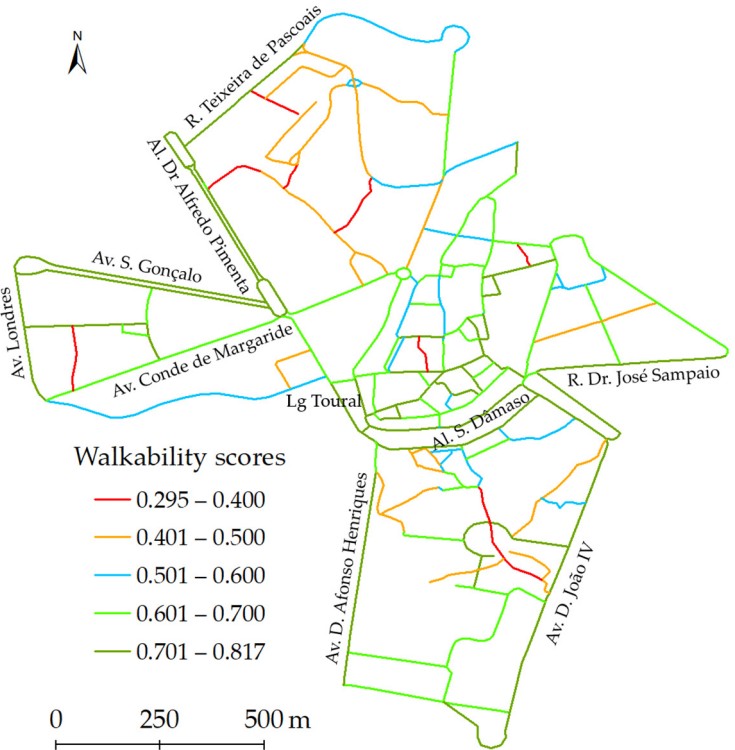

**Figure 4.** Evaluation of the walkable conditions according to the six criteria.

**Table 6.** Synthesis of the walkability assessment.

| Walkability Score Classes | Street Paths | | Street Length | |
|---|---|---|---|---|
| | By Class | Accumulated | By Class | Accumulated |
| <0.400 | 7.73% | 7.73% | 4.48% | 4.48% |
| 0.401–0.500 | 13.53% | 21.26% | 15.46% | 19.94% |
| 0.501–0.600 | 14.49% | 35.75% | 14.77% | 34.71% |
| 0.601–0.700 | 35.75% | 71.50% | 30.14% | 64.85% |
| >0.701 | 28.50% | 100.00% | 35.15% | 100.00% |

The analysis shows that the overall conditions provided by this pedestrian network are favourable. As shown in Table 6, 28.5% of the street paths are within the best-scored

class (>0.701), while around 65% of the streets' path lengths scored above 0.60. There were no street paths ranked below 0.29, but there were also no street paths which scored above 0.82. The best-ranked street paths can be mostly found in the city centre (Alameda S. Dâmaso, Largo do Toural, Rua Doutor Avelino Germano, Rua Alfredo Guimarães, and Rua da Rainha Dona Maria II, among others), and in some collector streets (Alameda Doutor Alfredo Pimenta, Avenida S. Gonçalo, Avenida de Londres, and Rua Dr. José Sampaio, among others). In turn, the streets that ranked worst are widespread in the study area, but with some prevalence in the north sector, which is mostly a residential area.

Besides the overall favourable conditions, the study area does not have a pedestrian network of high walkability. While at the scale of the city centre, there is a network of streets providing good levels of walkability, at the scale of the study area there are several discontinuities and interruptions in the network, and pedestrians may find changeable conditions in specific routes. This can make walking less comfortable and attractive and encourage the use of other modes of transport for short urban trips.

## 5. Discussion

Pedestrian networks are considered essential to an equitable and sustainable urban mobility system [5]. In this paper, we describe a GIS-based multi-criteria method transferred from the SPN research project to evaluate the walkability of a pedestrian network in Guimarães, Portugal. The evaluation was performed as a composite quality of the urban space produced by the combination of 19 built environment and streetscape attributes grouped into six main criteria. These attributes represent essential spatial requirements that make a pedestrian network convenient, comfortable, connected, safe, and attractive, and they may therefore be decisive for the decision and satisfaction of walking on a daily basis. The relative importance of these attributes was defined by a group of experts in urban/transport planning by using the AHP method. The contribution of this paper was to check the spatial transferability of the SPN method and to evaluate the level of walkability provided by the pedestrian network in the centre of Guimarães.

Regarding the first highlight, the spatial transferability reflects the capacity of a given measure to describe walkability. In this study, we demonstrated how the SPN method, which was firstly implemented in Bologna and Porto, can be transferred to other cities using entirely different local data sources and weights. We described how open and city datasets can be used to develop the evaluation, how to collect specific streetscape data through audits, how to evaluate the various sub-criteria, and the GIS operations carried out to map walkability. Regarding the SPN model, we only changed the weights assigned to the criteria and sub-criteria. Instead of using the average mean retrieved from a questionnaire administered in both cities, we decided to support the weighting process with a panel of experts from the University of Minho through an AHP. This change was made to make the evaluation more connected with the milieu of Guimarães and to avoid the perceptions of people from cities with different pedestrian conditions. The obtained scores differ from those of SPN [53]. Although there is a similar relative importance among some criteria (traffic safety and security, accessibility, streetscape design), there are some differences, especially in the evaluation of sidewalk facilities. Whilst in the current study the condition of sidewalks and the presence of unobstructed sidewalks were the most scored, in the SPN, the various sidewalk facilities' attributes (excepting slopes) were similarly weighted. These differences could be explained by the different methodology used to score the attributes but also, as emphasised by Taleai and Amiri [52], by the different views that experts and residents may have about walking. This is not surprising since the weights assigned to attributes with an influence on walking are substantially different even if we consider studies from the same countries [75,76] or even from the same cities [25,26]. Above all, multi-criteria are flexible tools that can been adjusted according to the view of the people involved and to the specificities of a given city. Apart from this, the spatial transferability proves to be effective and functional in mapping the levels of walkability in the centre of Guimarães, in identifying the most and least walkable areas, and in identifying the gaps

in the pedestrian network. Furthermore, the method proves to be adjustable in terms of weighting the attributes according to local specificities. These results indicate that the described method could be used in other cities to assess the walkability of a pedestrian network. This is an important methodological step towards the wider use of the described method on walkability research.

Regarding the second highlight, the method allowed the identification of different levels of walkability within this pedestrian network. The results demonstrate that the general conditions found in the study area can be classified as pedestrian-friendly. Around 64% of the street length scored above 0.60, a value that according to Moura et al. [25] was used to classify highly walkable areas. In part, these results reflect the overall good performance of the three criteria highly scored by the experts: sidewalk facilities, accessibility, and street connectivity; on the other hand, the criteria with the lowest performance (streetscape design and land use) were those lower-scored by the experts. Usually, the central areas of cities are more pedestrian-friendly than the remaining urban spaces [77], which may also explain the good performance obtained. However, the highly scored street paths are not arranged into a network that allows pedestrians to walk comfortably, safely, and pleasantly between specific locations within the study area. Besides the relatively small area analysed (1.6 km$^2$), there are street paths in poor condition, which may cause discomfort and make the walking trips less pleasant and convenient. In terms of performance, the street paths of the historic centre and in some collector streets were those with the highest scores and those more integrated. These findings corroborated previous research on historic cities, reporting that sidewalks along main streets usually have high walkability levels [78]. However, these results do not confirm that historic centres with compact urban structures and narrowed sidewalks are few pedestrian-friendly [79]. In Guimarães, the compact urban structure of the centre affected the performance of some criteria, such as the lack of street trees. However, the compact urban morphology does not constrain the performance of the remaining sidewalk attributes and contributes to the good performance of this area, for example, in terms of traffic safety and security. Our findings also indicate that the pedestrian network of Guimarães is mostly connected and convenient, which were the two fundamentals scored above 0.70. This could be explained by the high street connectivity and accessibility to diverse urban functions that reduce travel distances and make walking more convenient.

By integrating the mesoscale and microscale characteristics of the urban environment weighted by a panel of experts, this study also contributes to filling some of the identified research gaps, e.g., the lack of streetscape variables on walkability evaluations [19]. Our results led us to conclude that streetscape and sidewalk attributes are the most relevant elements for evaluating the levels of walkability in this pedestrian network. They determine the levels of comfort and pleasantness experienced by pedestrians and represent around 40% of the relative importance of all the criteria analysed. The condition of sidewalks and the presence of unobstructed sidewalks were found to be particularly relevant. This finding confirms previous research indicating that the condition of sidewalks and the lack of obstacles have a significant influence on pedestrian comfort and on preventing falls and accidents [9,35]. When compared to sidewalk facilities, streetscape design attributes were much less important, but from these, complexity emerged as the most relevant. This finding indicates that a variety of buildings shapes, sizes, materials, and colours, and a diversity of architectural styles, historical buildings, and ornamentation make urban environments more attractive for pedestrians. It also confirms previous research on the importance of these elements on walkability [66,80]. In terms of mesoscale attributes, accessibility was the most relevant criteria. Abundant research has shown that travel distance is critical for pedestrians [21,81] and that key employment and neighbourhood amenities, such as retail, educational, and health services should be provided within 15 or 20 min walking distances [29,82]. The overall good performance reached on this criterion reflects the compact urban morphology and the spread of urban functions by the study area, which reduces walking distances and makes walking more convenient. In turn, traffic safety

and security were found to be less relevant for walking, which is in line with the findings obtained in other Portuguese and European cities [40,53,77].

In terms of planning, there is wide recognition that planners and decision-makers should be engaged in making more inclusive and sustainable cities [61]. By assessing the conditions provided to pedestrians, the described method can help urban planners and decision-makers in identifying problems (lower-scored paths and interruptions in a pedestrian network) and in defining planning policies to mitigate such problems. More specifically, the walkability maps could be used to identify which sub-criteria/criteria are affecting the performance of a pedestrian network. This information can be then used to identify potentially high-leverage interventions and to prioritise actions and investments to improve the overall quality of pedestrian networks. Moreover, as the method is flexible, the weights and the sub-criteria can be adjusted to evaluate more appropriately local specificities.

Finally, this study has some limitations that should be addressed in future studies. First, the described findings result from the replication of the SPN model on the centre of Guimarães, a historic Portuguese medium-sized city. In the planning domain, it is often recommended that further research is necessary to understand the performance and applicability of specific methods in different urban settings [83,84]. Thus, it would be interesting to transfer and replicate this SPN methodology to understand its performance and applicability in cities with other structures, sizes, and transport systems. Second, the described method gives a global evaluation of the pedestrian conditions, which do not reflect the influence of specific neighbourhood and streetscape attributes on different walking trips and to the needs of specific pedestrian groups. For example, some studies have reported that utilitarian and recreational walking are substantially different in terms of frequency, speed, duration, location, and related built environment attributes [77,85]. The research of Moura et al. [25] also indicates that differentiating the analysis for different types of pedestrian groups (adults, seniors, and impaired pedestrians) does have a significant impact on the walkability evaluation. In the future, it would be interesting to study how the described model performs and can be adapted to explore the relationships between the built environment, different types of walking, and different pedestrian groups. Third, the method is supported on the 19 attributes used in the SPN project. Besides covering key pedestrian network fundamentals, there are other sub-criteria that may also influence walkability, such as traffic volume, pedestrian crossings, route directness, and street lighting. These attributes were not evaluated but could be important for describing walkability. Finally, the performance of the various sub-criteria was analysed through a dichotomous scoring method (1/0) reflecting the presence or absence of specific elements. Despite being commonly adopted in walkability studies [25,32,75], this binary evaluation may not be the most appropriate to use, since it may mask specific differences and details of the attributes. The use of intermediate scores could be more suitable to classify some attributes (poor, acceptable, good) in order to have a finer description of the conditions provided to pedestrians.

## 6. Conclusions

Pedestrian networks are an essential component for creating walkable cities in order to make urban mobility more equitable and sustainable. In this study, we evaluated the level of walkability in a pedestrian network in Guimarães, Portugal by using a GIS-based multi-criteria method transferred from the SPN research project. The SPN method was chosen for its comprehensive use of a wide range of neighbourhood and streetscape attributes for describing walkability in Bologna and Porto, two historic European cities.

In the author's view, this study makes various contributions towards the methodological background of walkability assessment tools. The obtained results seem to confirm the potential spatial transferability of this method to evaluate walkability in pedestrian networks of different cities. Furthermore, the method proved to be efficient in evaluating the level of walkability in the centre of Guimarães according to 19 built environment and

streetscape attributes. The method also proved to be flexible as the weights are adjustable and the sub-criteria may also be adapted to more appropriately reflect specific local conditions. Moreover, the data sources needed and the process to evaluate the performance of each attribute are relatively simple, though time-consuming for certain micro-scale attributes. In conclusion, this method offers researchers, planners, and decision-makers a comprehensive tool to evaluate walkability in pedestrian networks that can be used to develop more walkable cities as a way of creating more sustainable, affordable, and equitable urban communities.

**Author Contributions:** Conceptualization, F.F. and E.F.; methodology, E.F. and E.F.; software, F.F.; validation, F.F. and E.F.; formal analysis, E.F.; investigation, F.F.; resources, E.F.; data curation, E.F.; writing—original draft preparation, F.F.; writing—review and editing, E.F. and R.R.; visualization, F.F.; supervision, R.R.; project administration, R.R. All authors have read and agreed to the published version of the manuscript.

**Funding:** This research received no external funding.

**Institutional Review Board Statement:** Not applicable.

**Informed Consent Statement:** Not applicable.

**Data Availability Statement:** Not applicable.

**Acknowledgments:** The authors would like to thank the Centre for Territory, Environment and Construction, University of Minho for technical support.

**Conflicts of Interest:** The authors declare no conflict of interest.

## Appendix A

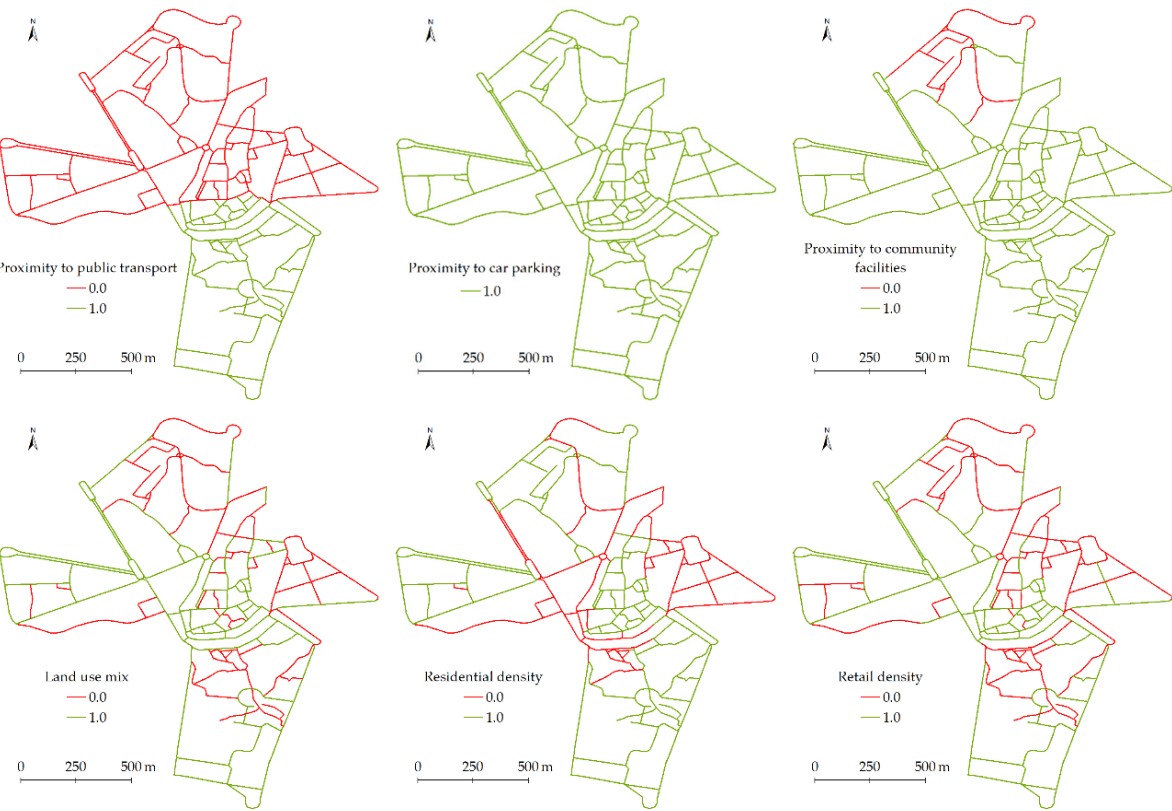

**Figure A1.** *Cont*.

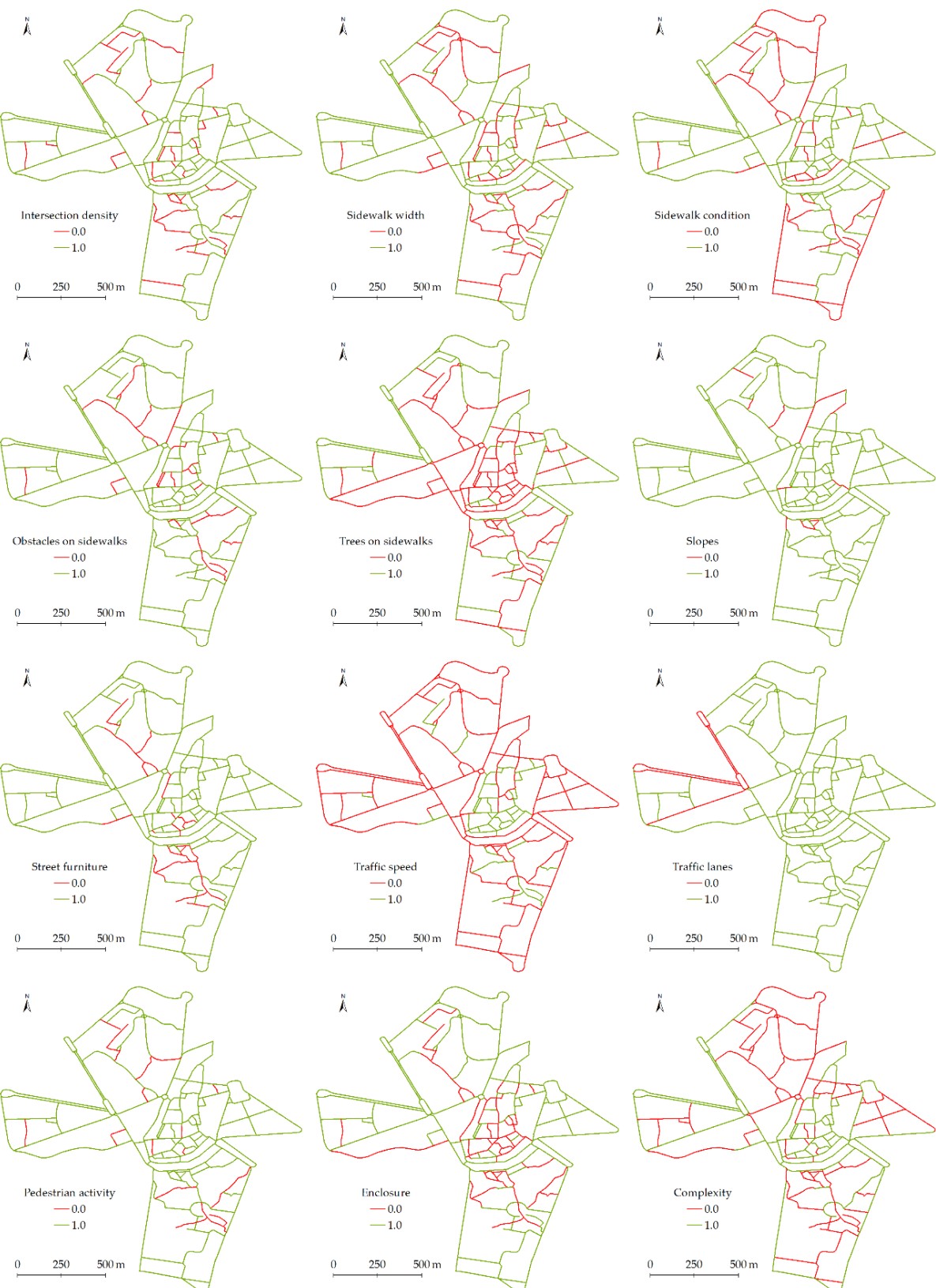

**Figure A1.** *Cont.*

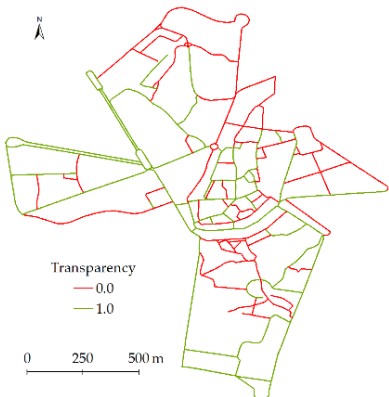

**Figure A1.** Evaluation of the 19 built environment and streetscape attributes in the study area.

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
