# Peer review of "Walkable Cities: Using the Smart Pedestrian Net Method for Evaluating a Pedestrian Network in Guimarães, Portugal"

_sustainability, doi:10.3390/su141610306_

Round 1

Reviewer 1 Report

Dear Authors, 

Have a nice day!

This paper is about walkability and its attributes to better conceptualise pedestrian networks and sustainable mobility in Guimaraes. The core contribution of this paper is its methodology. The audience of this work might be limited, but its implications are broad. Overall, I found it a very interesting piece of work and didn't find any major problems. However, there are some minor issues: 

1) The study area is Guimaraes, as mentioned in the title and in the abstract. However, lines 74-76 also refer to two other pedestrian networks in Bologna (Italy) and Porto (Portugal). It requires clarity. 

2) There is a mistake in numbering figures. Figures 1 and 2 are both presented as figure 1. 

3) Keywords provide a chance to broadcast our work, and repetition of keywords from the title makes us limit the accessibility of our work to other researchers. So, I suggest authors use unique keywords and not repeat those used in the title. 

Author Response

Please, find enclosed the author's response letter to Reviewer 1.

Reviewer 2 Report

The article is very well done. The article is writing in a very comprehensive and understandable way what makes me think that the written is good. In my opinion the article should be accepted in present form.

Good luck.

Author Response

Please, find enclosed the author's response letter to Reviewer 2.

Reviewer 3 Report

Thank you for the opportunity to read and review this article.

It is an excellent application of a multi-criteria attribute model for the walkable city. 

While the article is very robust and your indicators are very appropriate to what you want to measure, in the same way the estimation of the weights. 

However, perhaps the improvement that could be made, is the summary of the attributes point. 3.2.1 in a summary table.

Also, a process of calculating the AHP values for the area could be developed. 

Finally, at the end, as a discussion and conclusion, a model analysis should be incorporated that shows in more detail and that emphasises the (unnumbered?) carographs on lines 464 to 466, as well as the asymptotic one on line 541.

Without further ado, I think it's a great job, my sincere congratulations.

Author Response

Please, find enclosed the author's response letter to Reviewer 3.
